# A Meta-analysis Describing the Effects of the Essential oils Blend Agolin Ruminant on Performance, Rumen Fermentation and Methane Emissions in Dairy Cows [note 1]

**DOI:** 10.3390/ani10040620

**Published:** 2020-04-03

**Authors:** Alejandro Belanche, Charles J. Newbold, Diego P. Morgavi, Alex Bach, Beatrice Zweifel, David R. Yáñez-Ruiz

**Affiliations:** 1Estación Experimental del Zaidín, Consejo Superior de Investigaciones Científicas (CSIC), Profesor Albareda 1, 18008 Granada, Spain; david.yanez@eez.csic.es; 2Scotland’s Rural College (SRUC), Edinburg campus, Peter Wilson Building, King’s Buildings, Edinburgh EH9 3JG, UK; Jamie.Newbold@sruc.ac.uk; 3Université Clermont Auvergne, INRAE, VetAgro Sup, UMR 1213 Herbivores unit, F-63122 Saint-Genès-Champanelle, France; diego.morgavi@inrae.fr; 4Institució Catalana de Recerca i Estudis Avançats (ICREA), 08005 Barcelona, Spain; alex.bach@icrea.cat; 5Institut de Recerca i Tecnologia Agroalimentàries (IRTA), Department of Ruminant Production, 08140 Caldes de Montbui, Spain; 6Agolin SA, 1145 Bière, Switzerland; beatrice.zweifel@agolin.com

**Keywords:** dairy cows, essential oils, meta-analysis, methane, milk yield

## Abstract

**Simple Summary:**

Increasing feed efficiency and decreasing environmental impact are key targets in ruminant sciences. This meta-analysis suggested that supplementation of lactating dairy cows with the essential oil blend Agolin Ruminant^®^ (at 1g/d per cow) during a period greater than 4 weeks had a positive effect on milk yield (+4%) and decreased methane emissions (−10%) without affecting feed intake and milk composition. Although the mode of action is still unclear, this nutritional strategy seems to represent an encouraging alternative to improve productivity in commercial farms.

**Abstract:**

There is an increasing pressure to identify feed additives which increase productivity or decrease methane emissions. This paper aims to elucidate the effects of supplementing a specific essential oils blend Agolin^®^ Ruminant on the productivity of dairy cows in comparison to non-treated animals. A total of 23 in vivo studies were identified in which Agolin was supplemented at 1 g/d per cow; then a meta-analysis was performed to determine the response ratio on milk yield, rumen fermentation, methane emissions and health. Results indicated that an adaptation period of at least 4 weeks of treatment is required. Whereas short-term studies showed minor and inconsistent effects of Agolin, long-term studies (>4 weeks of treatment) revealed that Agolin supplementation increases milk yield (+3.6%), fat and protein corrected milk (+4.1%) and feed efficiency (+4.4%) without further changes in milk composition and feed intake. Long-term treatment also decreased methane production per day (−8.8%), per dry matter intake (−12.9%) and per fat and protein corrected milk yield (−9.9%) without changes in rumen fermentation pattern. In conclusion, despite the mode of action is still unclear and the small number of studies considered, these findings show that Agolin represents an encouraging alternative to improve productivity in dairy cows.

## 1. Introduction

Dairy production faces numerous global challenges, while milk demand is globally rising; increasing concerns appear over its environmental impact including methane (CH_4_) emissions, and the transfer of antibiotic resistance from animals to humans [1]. In response to these concerns more efficient and safe animal production systems need to be developed. One of the alternatives that has shown promising potential is the use of additives based on plant-extracted essential oils (EO) to manipulate rumen fermentation, boost animal productivity and decrease CH_4_ emissions [2]. However, the available information in the literature generally involves studies that are either in vitro or of a short duration in vivo without the possibility of recording production data consistently.

One of the main drawbacks of EO feed additives is that the rumen microbial ecosystem can adapt and revert the effect, which results mostly in a short-term benefit (i.e., improved microbial fermentation or lower CH_4_ production per day). For example, the use of EO or their active ingredients have shown positive results on reducing CH_4_ emissions in in vitro batch cultures, whereas no or only a transient effect on the rumen fermentation pattern was found in continuous cultures or in vivo [3,4]. For example, Cardozo et al. [5] reported positive short-term effect of various EO on the in vitro fermentation characteristics but most of them disappeared after 6 days. This indicates that microbial adaptation can occur in vitro and highlights the need to conduct in vivo studies that run for long enough time to detect such effects.

Although the use of EO to enhance productivity and decrease enteric CH_4_ emissions is not new [1,4], the few in vivo studies which supplemented EO to ruminants reported highly variable responses due to a number of reasons including the type and dose of EO used, the proportion of each EO within the mixture, the stage of lactation, production level and the type of diet fed to the animals, among many others [2]. In addition, availability of representative production data is limited if only published research conducted in experimental conditions is considered without including data recorded from commercial dairy farms, aspect that could bias the results on-farm conditions.

Agolin^®^ Ruminant (Agolin SA, Bière, Switzerland) is a commercially available pre-mixture of flavorings. The main active compounds of this product are food grade and chemically-defined plant extracts including coriander (*Coriandrum sativum*) seed oil (up to 10%), eugenol (up to 7%), geranyl acetate (up to 7%) and geraniol (up to 6%) along with some preservatives such as fumaric acid. Various authors have recently studied the effect of Agolin Ruminant on rumen fermentation in vitro [6,7,8] and in vivo [8,9,10,11,12,13,14,15,16] showing positive but variable results. As the product is in the market, numerous on-farm studies with dairy cows have been conducted in different countries over recent years, although the results have not been generally published in peer reviewed journals. The objective of the present meta-analysis was to quantitatively summarize the effect of Agolin^®^ Ruminant on dairy cows’ performance, physiology and enteric CH_4_ emissions using both animal studies under experimental conditions and on commercial dairy farms to describe its short- and long-term effects.

## 2. Materials and Methods

### 2.1. Literature Search and Study Description

To perform a robust meta-analysis, the Preferred Reporting Items for Systematic reviews and Meta-Analyses (PRISMA) guidelines was used to identify the studies [17] (Figure 1). Briefly, the server of the CSIC (Granada, Spain) was used to conduct an electronic literature search including CAB database, Medline, PubMed, Sciences-Direct and Web of Sciences. There was no restriction to peer-reviewed journals, and the eligible publications included abstracts, conference proceedings and theses. In addition, different groups of investigators from Aberystwyth University (Aberystwyth, UK), INRAE (Clermont-Ferrand, France) and IRTA (Caldes de Montbui, Spain) were contacted asking for unpublished studies. Since Agolin Ruminant has been in the market for several years, Agolin SA put us in contact with farms that have used the product to seek information from unpublished on-farm studies. In order to make results comparable, only those studies which met the following selection criteria were included in the meta-analysis: (1) Only in vivo studies using dairy cows were considered, (2) For any variable, studies had to report mean value and its variability for both treated (AGO) and control groups (CON), 3) Only studies which used the EO blend Agolin® Ruminant at the recommended dosage of 1 g/d per cow were considered, and 4) On-farm studies which did not collect data systematically were not considered. For example, if animals were not randomly distributed across treatments, if the diet or EO dosage changed without justification during the course of the study, or if relevant data such as milk yield were not reported or measured just once.

From the initial 29 studies overviewed, a total of 23 studies satisfied the selection criteria and were included in the meta-analysis (Figure 1). All these studies were conducted during the last decade across 10 different countries and were based on intensive dairy production system using Holstein-Friesian cows, except study nº 21 which used Simmental-Montbéliarde cows (Table 1). Multiparous cows were used in 9 studies (primiparous in 3) while the rest of studies contained a mix of both types of animals. All studies used total mixed rations (TMR) containing a mix of various forages (e.g., corn and grass silages, grass and alfalfa hays, whole crop or straw) representing between 52% and 83% of the diet. The concentrate was generally composed of a mixture of cereals, soybean meal and protein blends. Although all studies provided information about the AGO and CON treatments, longitudinal experimental designs varied across studies: 3 studies used a crossover design in which all animals passed through both experimental treatments, 8 studies consisted in randomized complete block design in which two groups of animals ran in parallel, and 12 studies had a straight through design in which a single group of animals was studied before and after receiving the experimental treatment. Thirteen studies (most of them conducted in research facilities) reported data from individual cows, thus the animal was considered the experimental unit. On the contrary, ten farm studies had a large number of animals (350 ± 277 cows per treatment) but the pen was considered the experimental unit due to the lack of individual animal data, aspect that decreased substantially the number of replicates (6 ± 1.3 replicates per treatment). In terms of feeding, the EO blend was incorporated into the ration by either top-dressing or as part of the TMR.

### 2.2. Data Extraction

The majority of the studies (Table 2) provided information about milk yield and composition, dry matter intake (DMI) and feed conversion efficiency (FCE). Overall a high daily milk yield was observed (30.77 ± 6.48 kg/cow) across studies although with substantial differences depending on the parity number and the number of days in milk (DIM). Rumen fermentation data and CH_4_ emissions were reported in 8 studies (7 for VFA molar proportion) under controlled experimental conditions. The production of CH_4_ was measured using various methodologies including respirometry chambers (studies 2, 10 and 13), sulfur hexafluoride “SF6” tracer technique (studies 14 and 15), Green-Feed system (C-Lock Inc., Rapid City, SD, USA. study 4), and in vitro incubations (studies 5 and 6). These two later studies were included because the in vitro incubations were part of large in vivo studies in which animals were adapted to the additive when used as donors. Rumen fluids were incubated at 39 °C for 24h without substrate (study 6) or for 48 h with substrate (study 5) before measuring CH_4_ concentration in the headspace by gas chromatography as previously reported [18] To avoid a bias that might be introduced due to the different methods used to measure CH_4_, the effect of the EO supplementation was expressed as a change with respect to the control. Despite this consideration, special care should be taken given the small number of CH_4_ observations. Fat and protein corrected milk yield (FPCM) was calculated for 4.0% fat and 3.3% protein content [19]. Fat and protein corrected milk yield (FPCM) was calculated for 4.0% fat and 3.3% protein content [19]. When a straight through experimental design was used, FPCM yield was also adjusted for the DIM according to the milk yield persistency described for primiparous and multiparous cows [20]. Mean somatic cells counts (SCC) and rumen protozoa concentrations were log transformed to attain normality. If a study reported production for primiparous and multiparous cows separately, data were considered as independent studies. Only those parameters which were reported by at least three studies were included in the meta-analysis. A funnel plot analysis was performed to test the asymmetry across studies and to discard potential bias between published and unpublished studies and between different experimental designs considering the milk yield as the key response factor.

### 2.3. Meta-analyses

The aim of the meta-analysis was to provide simple, broad scale parameters estimates to describe the overall effects of supplementing a commercial EO blend (Agolin Ruminant^®^) on dairy cows productivity. The effect of Agolin supplementation on the studied parameters was evaluated using the effect size method [21] which allows the comparison of two populations. For each parameter, the effect size based on means was calculated as the response ratio (*R*) as follow:*R_i_* = *A_i_/C_i_*(1)

Where *R_i_* is the observed response ratio, *A_i_* is the reported mean for the AGO treatment and *C_i_* is the reported mean for the CON treatment in the *ith* study. This approach allowed the outcome of different studies to be expressed on a common scale [22]. For statistical purposes the response ratio was further transformed to natural-logarithm to achieve normality.

The precision of the estimate was based on the number of observations (*n*) and the reported standard deviation (SD) of the CON and AGO groups or the standard error of the mean (SEM). When SEM was not reported, it was calculated by dividing SD by square root of *n*. When a calculation of SD was not possible, it was estimated as the pooled SD from all the other available studies included in the meta-analysis [23]. Then, the variance of the response was calculated as follows:*V_i_* = *S_i_*^2^ × ((1/ *nA_i_* × *A_i_*^2^) + (1/*nC_i_* × *C_i_*^2^))(2)
where *V_i_* is the estimated variance of *Ri*, *S_i_*^2^ is the pooled standard deviation, *nA_i_* is the sample size of AGO group and *nC_i_* is the sample size of the CON group of the *ith* study. Thus, this method weighted the studies according to the number of observations (*n*). 

Meta-analysis was performed using the Metafor package (Version 2.1.0) from R statistics (Version 3.6.1). In particular, the function rma was used since provides a general framework for fitting meta-analytic models. This meta-analysis was based on studies that were not exactly identical in the type of animals (e.g., breed, lactation number, DIM), type of diet, experimental design or treatment duration. These differences between studies may introduce variability (“heterogeneity”) among the true effects. However, considering the uneven distribution of these factors and the limited number of studies included in this meta-analysis, a random model was used to maximize the degrees of freedom and ultimately the statistical power. This approach implied the existence of between-studies variance based on the assumption that studies included in the analysis correspond to a random sample of all possible studies [24]. The meta-analysis was calculated fitting a random-effect model with a DerSimonian-Laird estimator [25] for assessing heterogeneity (*τ*^2^) for each category separately as follow:
*Y_i_* = *R* + *R_i_* + *e_i_*(3)
where *Y_i_* is the true response ratio in the *ith* study, *R* is the overall true response ratio mean, *R_i_* is the random deviation of the *ith* study from the overall response ratio calculated from the variance described above [*R_i_ ~ N (0, τ^2^)]* while *e_i_* is the random error [*e_i_ ~ N (0, V^2^)]* [22]. Forest plots were generated to illustrate the response ratio along with the estimated 95% confidence interval and sample size for each parameter considered (Appendix A). 

The heterogeneity or between-studies variability was determined using the Cochran’s *Q* test as indicator of the inconsistency across studies. However, as the *Q* statistics does not provide information on the extent of true heterogeneity (only its significance), the *I*^2^ statistics was also calculated, which denotes the percentage of the total variability that is attributed to the between-studies variability. [26]. For the response ratio, values below 1 indicate a negative, while values above 1 indicate a positive effect of EO supplementation on that particular parameter. Significant effects were declared at *p* < 0.05, whereas observations were considered inconsistent when heterogeneity tests indicated *I*^2^ > 50 and *Q* < 0.05 [26].

### 2.4. Statistical Analyses

A general meta-analysis was performed capturing the information across all studies independently of the treatment duration with AGO in order to maximize the number of parameters and observations. However, most EO blends require an adaptation period to express its maximum positive effects on animal productivity [27]. Thus, those studies which reported the FPCM yield progress (as the main parameter of interest) after Agolin supplementation were selected to determine the length of the adaptation period. Then, the FPCM yield increment in the AGO respect to the CON group was calculated for each week based on a repeated measures analysis using the SPSS software (IBM Corp., Version 21.0, Armonk, NY, USA).

Since the repeated measures analysis revealed a substantial increase in FPCM after 4 weeks of treatment with EO, it was decided to conduct two more meta-analyses for the most relevant parameters. The first meta-analysis compiled short-term data (<28 days of Agolin treatment) derived from short-term studies and from the first 28 days of long-term studies. This short-term meta-analysis reported milk yield (19 studies), milk composition (10), rumen fermentation (7) and CH_4_ emissions (9 studies). The second meta-analysis evaluated the long-term information (>28 days of Agolin treatment) in terms of milk yield (19 studies), milk composition (9), rumen fermentation (4) and CH_4_ emissions (7 studies).

## 3. Results

### 3.1. Overall Effects of Agolin

The funnel plot showed a symmetrical distribution across studies for milk yield indicating no bias (Egger bias *p* > 0.10) between published and unpublished studies (Figure 2A) or between different experimental designs (Figure 2B). Although 83% of the studies fell within the range of 2 standard deviations of the mean, results from straight through studies tended to have greater variability than randomized block designs. The overall meta-analysis considering the entire duration of the treatment (Table 3 and Appendix A) showed that Agolin supplementation increased milk yield (*R* = 1.020), FPCM yield (*R* = 1.031) and FCE (*R* = 1.030) with a low inconsistency across studies. On the contrary, DMI and milk composition was unaffected by the EO supplementation. Rumen fermentation pattern, in terms of pH, concentrations of total VFA, rumen protozoa and VFA molar proportions, was not significantly affected by Agolin supplementation. Agolin supplementation decreased the CH_4_ production (g/d, *R* = 0.954, *p* = 0.007) and intensity (g/kg FPCM, *R* = 0.925, *p* = 0.023) with a low level of inconsistency (*I*^2^ < 23, *Q* > 0.24). This decrease in CH_4_ emission was not significant when expressed as CH_4_ yield (g/kg DMI).

### 3.2. Short- and Long-term Effects of Agolin

The analysis of the weekly FPCM yield after AGO supplementation (Figure 3) showed a progressive increase over time (*p* = 0.003) which was only significantly higher than the CON treatment after 4 weeks of continuous supplementation. As the experiments included in the overall meta-analysis widely varied in the duration of Agolin treatment (from 22 to 365 days), it was decided to conduct two meta-analysis to describe the short-(<28 days of treatment) and long-term effects (>28 days of treatment) of Agolin supplementation.

The meta-analysis addressing short-term exposure found no significant effects of AGO supplementation in respect to the CON (Table 4 and Appendix A) for most of the parameters analyzed. No differences were noted for DMI, milk composition, rumen fermentation and FCE (*p* > 0.1). Agolin addition slightly increased milk yield (*R* = 1.026) and FPCM yield (*R* = 1.028) but with an extremely high degree of heterogeneity across studies (*I*^2^ > 69, Q < 0.001). The short-term treatment with Agolin slightly decreased CH_4_ production (*R* = 0.978) and intensity (*R* = 0.974) but not CH_4_ yield.

The meta-analysis dealing with long-term treatments showed more significant effects (Table 5 and Appendix A). As described before, DMI, milk composition and rumen fermentation parameters were not substantially affected by a long-term treatment with Agolin. Milk yield (*R* = 1.036) and FCE (*R* = 1.044) were positively affected by Agolin but with a high level of inconsistency across studies (*I*^2^ > 73, Q < 0.001). However, long-term treatment with Agolin increased FPCM yield (*R* = 1.041) and decreased CH_4_ production (*R* = 0.912), yield (*R* = 0.871) and intensity (*R* = 0.901) with a high consistency across treatments (*I*^2^ < 5, *Q* > 0.39).

## 4. Discussion

### 4.1. Animal Performance

This meta-analysis showed that DMI was not affected by Agolin supplementation, possibly because in 5 studies the TMR was offered at 95% of *ad libitum* to ensure uptake. Previous studies also reported a lack of effect of similar EO blends on DMI in dairy cows [28,29]. Regarding milk yield, several studies using a limited number of animals (4 cows) during a short treatment period (less than 4 weeks) have shown no differences in milk yield and milk components when cows are supplemented with EO blends [28,30] or plants rich in EO [29]. Our meta-analysis compiling information of studies with a variable duration (from 22 to 365 days of treatment with Agolin) and a larger number of animals showed a small, but consistent, increase in milk yield (+ 2.0%) without major changes in milk composition and SCC. Most studies conducted in mid-lactating cows agreed that Agolin had a positive effect on milk yield [11,12,13,15], one study with a large number of early-lactating cows reported no effect on milk yield but increased milk fat percentage [10], while two studies in late-lactation noted a small decrease in milk yield but associated to an increase in milk fat percentage [8,16]. The use of energy corrected milk (ECM) and FPCM yield are gaining more attention because they integrate information concerning milk yield and milk composition, thus normalizing data across studies. When productivity was expressed as FPCM yield, it was noted that Agolin promoted a greater increase in productivity (+ 3.1%) together with a greater consistency across studies, possibly as a result of the marginal but positive effect of the EO on milk fat and protein percentages. Regarding EO and feed efficiency, results are inconsistent in the literature. Some studies using other EO have reported increases in feed efficiency [31,32], but others have found no differences [4,29]. This inconsistency across studies may be due to variation associated to stage of lactation, type and combination of plant extracts, or dose used. To prevent this, in our study FCE was calculated as the ratio between the FPCM yield and DMI using the same EO blend and dose across studies. Results showed that Agolin supplementation increased FCE in the same order of magnitude than FPCM yield (+ 3.0%) but with a slightly greater heterogeneity given the inherent difficulty of measuring DMI on-farm conditions.

Several in vitro studies have suggested positive but transitory effects of EO on the rumen microbial fermentation [5,33,34]. The in vivo results presented here disagree with this hypothesis and demonstrated that dairy cows required an adaptation period to Agolin of at least 4 weeks to achieve consistent positive effects on milk yield. As a result, the short-term meta-analysis showed that Agolin promoted a small increase in milk yield (+ 2.6%) and FPCM yield (+ 3.6%) but with a low level of reproducibility. Although the effects of EO are highly dependent on the type of active compound and dosage [2], these findings could help to understand the inconsistent results observed in previous studies in relation to the use of EO to enhance milk yield in short-term studies [35].

The meta-analysis compiling experiments with long-term treatment with Agolin showed a different picture indicating a substantial increase FPCM yield (+ 4.1%) together with a high reproducibility of the results. Agolin also promoted similar increases in milk yield (+ 3.6%) and FCE (+ 4.4%) but with a greater level heterogeneity as a result of marginal changes in milk composition, difficulty to measure individual DMI and the limited number of studies reporting FCE. These findings suggest that Agolin promoted more efficient nutrient utilization by the animal. However, the specific mechanisms behind such effect are still to be elucidated. Enhanced digestion and absorption of metabolites has been reported in monogastrics fed EO [36], whereas in ruminants, the most plausible mechanisms involve the modulation of the rumen microbial fermentation [35].

### 4.2. Rumen Fermentation

Antimicrobial activity of EO have been demonstrated against a wide variety of microbes, thus it has been hypothesized that using EO may represent a valid strategy for rumen microbial modulation [37]. Agolin is an EO blend containing coriander seed oil, eugenol, geranyl acetate and geraniol as the main active compounds. Coriander oil (*Coriandrum sativum*) contains coriandrol and geranyl acetate which, together with geraniol, are monoterpenes with antibacterial, antifungal, antiprotozoal and anti-oxidant activities [37]. Eugenol is a phenolic present in clove bud (*Eugenia caryophyllus* or *Syzygium aromaticum*) [27]. In vitro batch culture studies have shown positive rumen modulatory effects of coriander oil, eugenol [33], geraniol [7] and their blend Agolin on the rumen fermentation [6,7], suggesting a theoretical increase in the efficiency of energy and protein metabolism in the rumen [34]. This hypothesis is based on the negative effects of EO on the rumen hyperammonia-producing bacteria and the positive effect on the propionate-producing bacteria [35]. However, our meta-analysis indicated that Agolin supplementation to dairy cows did not promote consistent changes in the rumen fermentation, in terms of total VFA concentration and molar proportions, as previously shown with other EO blends [4,35]. Former studies have generally not been able to reproduce the positive effects of EO on the in vitro fermentation when dairy cows have been supplemented with coriander oil [38] or eugenol [39]. These observations suggest that in vitro experiments do not always represent what occurs *in vivo*. Lack of change in VFA concentration could be viewed as desirable when is accompanied by positive effects on animal performance or decreased CH_4_ emissions as revealed by our meta-analysis. These results are in line with a recent study which reported no changes in the rumen fermentation when dairy cows were supplemented with coriander oil for 63 days while milk yield, nutrient digestibility and feed efficiency increased [38]. Our observations suggest that the small shift in rumen fermentation pattern is not likely to fully explain the positive effects of Agolin on milk yield. However, further studies to investigate the impact of Agolin supplementation on rumen function and microbial composition are needed.

### 4.3. Methane Emissions

Given the growing worldwide interest in decreasing CH_4_ emissions from domestic ruminants, the antimicrobial activity of EO has prompted interest in whether these compounds could be used to decrease rumen methanogenesis. Several studies have showed that EO can decrease CH_4_ production in vitro [35], however the challenge consists in finding a combination of EO that reduces CH_4_ production without a concomitant decrease in feed intake and productivity. Few studies have evaluated the effects of EO and their constituents on CH_4_ emissions in vivo [3,8,13,16], and only one study has assessed long-term effect of EO on rumen methanogenesis [16]. Our meta-analysis showed that Agolin is an EO blend that leads to a decrease in CH_4_ emissions, with the magnitude and consistency of this reduction dependent on the treatment duration. When short-term studies were considered, the CH_4_ inhibitory effect of Agolin was small (−2.3%) and largely inconsistent. On the contrary, when long-term studies were considered the anti-methanogenic effect of Agolin was consistent across studies leading to a decrease in CH_4_ production (−8.8%), CH_4_ yield (−12.9%) and CH_4_ intensity (−9.9%) without compromising feed digestibility or milk yield.

These findings should be cautiously interpreted given the relatively low number of studies included in the meta-analysis which reported CH_4_ emissions and the inherent differences across the measuring methods, including in vitro incubations. However it seems clear that there is a potential to further consider this EO blend as a CH_4_ mitigation strategy. There are several hypotheses to describe the anti-methanogenic effect depending on the EO considered such as the inhibition of certain rumen bacteria, methanogenic archaea, protozoa or shifting the rumen fermentation pattern [27]. This study was not able to identify a sole factor to explain the observed decrease in rumen methanogenesis; instead it provided insight about potential drivers which suggest a multi-factorial mode of action. The effect of EO blends on the rumen fermentation may be diet-dependent since an increase in total VFA has been reported in lactating cows fed alfalfa silage, while the opposite was true when they were fed corn silage [30]. In the present meta-analysis Agolin supplementation led to a numerical decrease in the total VFA concentration (−2.2%) which was in line with the small increase in rumen pH (from 6.58 to 6.64), possibly because corn silage was the most commonly used forage across studies. This decrease in total VFA, together with the small shift in the VFA proportions (more propionate and less butyrate) would lead to a theoretical 4% decrease in H_2_ production based on the VFA stoichiometry [40], explaining about half of the observed decrease in CH_4_ emissions. Cabezas-Garcia et al. [41] after conducting a meta-analysis concluded that digesta passage rate was one of the key driving factors explaining the between-cow variation in CH_4_ production, thus further research is needed to investigate the potential effect of this EO blend on digesta passage rate.

Previous analyses of the factors which determine rumen methanogenesis found an equation (CH_4_ in g/kg DMI = -30.7 + 8.14 × protozoa in log cells/ml) which correlates rumen protozoa and CH_4_ emissions [42]. This link seems to rely on the protozoal capacity to digests fibrous components into H_2_ and butyrate [43], as well as on the efficient interspecies H_2_ transfer towards their endo- and epi-symbiotic methanogens [44]. A more specific meta-analysis [37] noted the changes in CH_4_ production induced by EO supplementation were linearly associated with the protozoal numbers suggesting a 0.45% reduction in CH_4_ by each 1% decrease in protozoal numbers. Our meta-analysis agrees with these calculations since Agolin supplementation numerically decrease the rumen protozoal concentration (from 5.42 to 5.35 log cells/ml) which represented a 15% drop in the protozoal cells/ml, and therefore could explain about half of the observed decrease in CH_4_ emissions. Possibly the large variability in terms of diets and sampling times across studies precluded these differences to reach the statistical signification level. In a recent meta-analysis [43] it was demonstrated that elimination of rumen protozoa not only decreases butyrate molar proportion (−21%) and CH_4_ emissions (−11%) but also increases the flow of duodenal microbial N (+30%). Thus, microbial growth could act as alternative H sink and partially explain the observed decrease in methane intensity (g/kg FPCM) noted in this study.

## 5. Conclusions

This meta-analysis combining 23 experimental and farm studies across 10 different countries indicated that supplementation of lactating dairy cows with the essential oils blend Agolin Ruminant^®^ (at 1g/d per cow) exerted positive effects on milk production whereas it decreased enteric methane emissions in comparison to un-supplemented cows. These effects mostly appeared after an adaptation period of approximately 4 weeks of treatment and consisted in an increase in fat and protein corrected milk suggesting an improved feed utilization. These observations should be carefully interpreted due to the small number of studies available. Moreover, given that the specific mode of action involved in these effects is still unclear, further studies are needed to investigate the impact of Agolin on the rumen microbiome.

## Figures and Tables

**Figure 1 animals-10-00620-f001:**
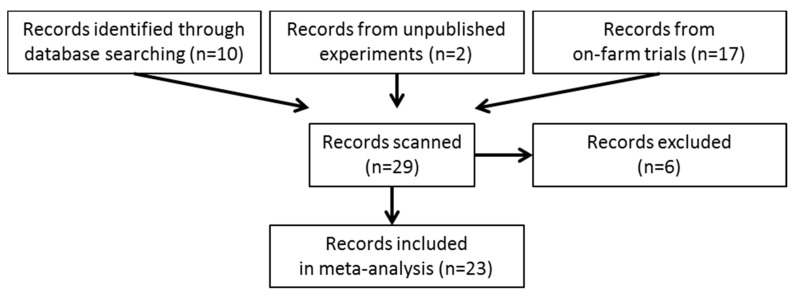
PRISMA flow diagram of all of the records searched and included in the meta-analysis.

**Figure 2 animals-10-00620-f002:**
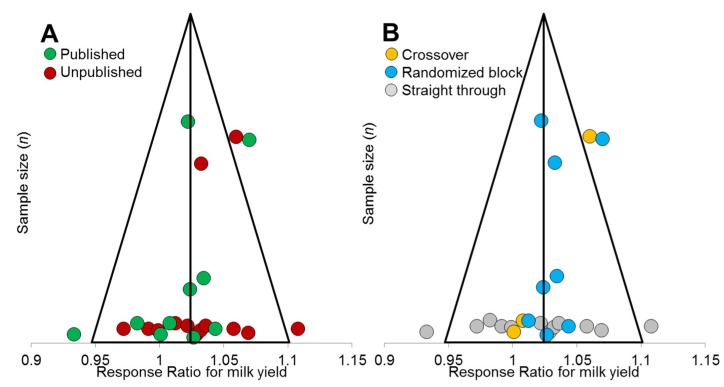
Funnel plot for all studies included in the meta-analysis of milk yield (*n* = 23) in order to detect bias between published and unpublished studies (**A**) or between experimental designs (**B**). Triangles represent the median Response Ratio (*R*) and a range equivalent to two standard deviations. If no publication bias is present, the data-points will be organized symmetrically.

**Figure 3 animals-10-00620-f003:**
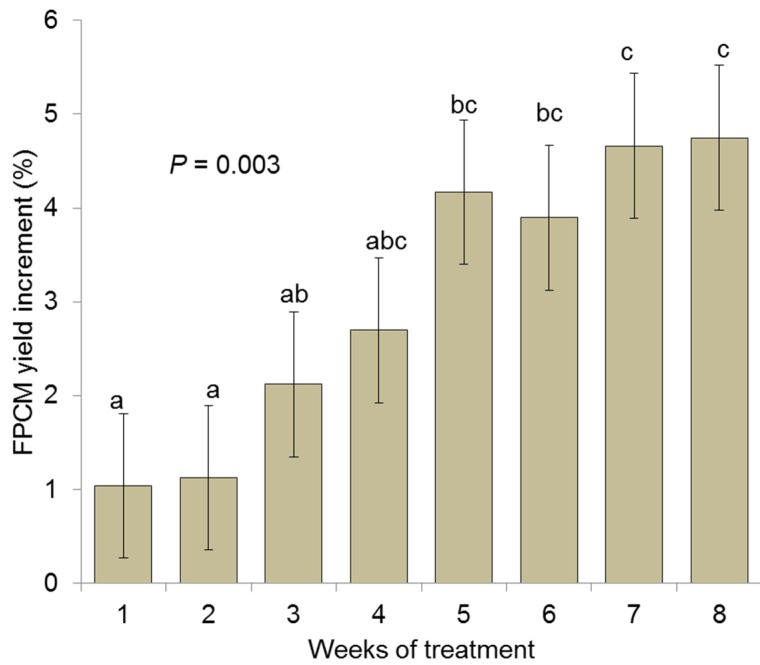
Effect of supplementing an essential oil blend (1 g/d per cow) to lactating dairy cows on the weekly fat and protein corrected milk yield (FPCM) based on 15 studies. The effect was expressed as a percentage of change respect to non-supplemented cows.

**Table 1 animals-10-00620-t001:** Summary of the studies included in the meta-analysis.

ID	Country	Year	Parity	Design	Unit	*n* ^1^	Days ^2^	Diet Ingredients ^3^	F:C ratio ^4^	Reference
1	USA	2009	Multiparous	Crossover	Pen	4	28	AH, WS, FA, ST, C	67/33	[10]
2	UK	2015	All	Crossover	Cow	8	35	GS, CS, PB, C	78/22	[14]
3	Hungary	2008	All	Crossover	Cow	76	28	CS, AH, SBM, C	55/45	Unpublished
4	UK	2016	Multiparous	Randomized block	Cow	75	174	GS, CS, PB, C	76/24	[13]
5	Netherlands	2017	2^nd^ parity	Randomized block	Cow	3	22	CS, GS, SBM, C	70/30	[9]
6	Spain	2015	Primiparous	Randomized block	Cow	24	56	GH, CS, AH, ST, SBM, C	60/40	[12]
7	Spain	2015	Multiparous	Randomized block	Cow	6	56	GH, CS, AH, ST, SBM, C	60/40	[12]
8	Spain	2016	All	Randomized block	Cow	20	56	GH, CS, AH, ST, SBM, C	80/20	[15]
9	Switzerland	2012	All	Randomized block	Cow	80	180	GS, CS, GH, C	67/33	[11]
10	Germany	2012	All	Randomized block	Pen	8	60	CS, GS, AS, SBM, C	52/48	Unpublished
11	Hungary	2010	All	Randomized block	Cow	65	92	CS, AH, SBM, C	55/45	Unpublished
12	Netherlands	2017	All	Straight through	Cow	8	70	CS, GS, SBM, C	70/30	[16]
13	Belgium	2011	Multiparous	Straight through	Cow	4	42	GS, CS, SBM, C	83/17	[8]
14	France	2011	Multiparous	Straight through	Cow	6	42	CS, GH, SBM, C	70/30	Unpublished
15	France	2014	Multiparous	Straight through	Cow	6	42	GS, GH, SBM, C	55/45	Unpublished
16	UK	2014	All	Straight through	Pen	5	30	GS, WB, PB, C	67/33	Unpublished
17	UK	2014	All	Straight through	Pen	6	53	GS, WW, PB, C	72/28	Unpublished
18	UK	2014	All	Straight through	Pen	6	57	GS, CS, PB, C	64/36	Unpublished
19	UK	2014	Primiparous	Straight through	Pen	6	244	GS, WW, ST, SBM, C	64/36	Unpublished
20	UK	2014	Multiparous	Straight through	Pen	6	244	GS, WW, ST, SBM, C	64/36	Unpublished
21	Italy	2017	All	Straight through	Pen	4	365	CS, GS, C	67/33	Unpublished
22	Spain	2016	Multiparous	Straight through	Pen	7	365	CS, GS, GH, C	70/30	Unpublished
23	Spain	2016	Primiparous	Straight through	Pen	7	365	CS, GS, GH, C	70/30	Unpublished

^1^ Experimental units. ^2^ Treatment duration. ^3^ AH, alfalfa hay; C, concentrate; CS, corn silage; FA, fresh alfalfa; GH, grass hay; GS, grass silage; PB, protein blend; SBM, soybean meal; ST, straw; WB, whole crop barley; WS, wheat silage; WW, whole crop wheat; ^4^ F:C ratio, forage to concentrate ratio

**Table 2 animals-10-00620-t002:** Descriptive statistics of the parameters included in the meta-analysis.

Parameter ^1^	Studies	Minimum	Maximum	Mean	Median	SD
Treatment duration (d)	22	22.0	427	143	80.5	125.0
Days in milk (d)	14	20.0	296	171	183	60.16
DMI (kg/d)	16	15.6	27.4	21.4	22.4	3.547
Milk yield (kg/d)	23	18.2	49.2	31.0	30.1	6.559
Milk Fat (%)	16	3.32	4.80	4.03	3.92	0.445
Milk protein (%)	16	2.79	3.51	3.25	3.29	0.190
Milk lactose (%)	8	4.43	5.27	4.75	4.76	0.206
Milk SCC (log/mL)	3	3.91	4.92	4.46	4.63	0.429
FPCM yield (kg/d)	20	21.3	47.1	32.9	32.1	5.957
FCE (kg/kg)	16	1.17	1.97	1.52	1.48	0.207
Rumen pH	3	6.46	6.78	6.62	6.62	0.147
Total VFA (mmol/L)	8	50.8	165	103	101	29.56
Acetate (%)	7	57.7	76.5	66.3	66.8	6.252
Propionate (%)	7	14.9	26.0	19.3	18.4	3.630
Butyrate (%)	7	8.74	14.1	10.9	10.0	1.931
Protozoa (log cells/mL)	3	5.00	5.80	5.43	5.52	0.333
CH_4_ production (g/d)	8	229	445	321	291	78.86
CH_4_ yield (g/kg DMI)	8	9.79	46.2	19.7	17.0	10.75
CH_4_ intensity (g/kg FPCM)	8	6.66	17.2	12.2	13.3	3.310

^1^ DMI, dry matter intake; SCC, somatic cell counts; FPCM, fat and protein corrected milk; FCE, feed conversion efficiency; VFA, volatile fatty acids.

**Table 3 animals-10-00620-t003:** Mean response ratio and estimated heterogeneity parameters describing the overall effects of supplementing an essential oil blend (1 g/d per cow) to dairy cows.

Parameter ^1^	*n*	Response Ratio (*R*)	95% CI	*p*-Value	Heterogeneity	
			Min. Max.		*I* ^2^	*Q*
DMI (kg/d)	16	1.003	0.985-1.020	0.737	86	<0.001
Milk yield (kg/d)	23	1.020	1.011-1.028	<0.001	16	0.248
Milk Fat (g/d)	16	1.004	0.979-1.029	0.739	85	0.000
Milk protein (g/kg)	16	1.002	0.996-1.008	0.419	46	0.023
Milk lactose (g/kg)	8	0.998	0.992-1.003	0.519	76	<0.001
Milk SCC (log/mL)	3	0.994	0.944-1.045	0.800	69	0.040
FPCM yield (kg/d)	20	1.031	1.026-1.035	<0.001	0	0.995
FCE (kg/kg)	16	1.030	1.011-1.049	0.002	34	0.087
Rumen pH	3	1.006	0.989-1.022	0.476	0	0.511
Total VFA (mmol/L)	8	0.982	0.946-1.019	0.346	0	0.685
Acetate (%)	7	1.002	0.991-1.011	0.756	91	<0.001
Propionate (%)	7	1.011	0.945-1.082	0.744	97	<0.001
Butyrate (%)	7	0.991	0.963-1.019	0.525	7	0.377
Protozoa (log cells/mL)	3	0.977	0.924-1.032	0.405	39	0.193
CH_4_ production (g/d)	8	0.954	0.921-0.987	0.007	23	0.241
CH_4_ yield (g/kg DMI)	8	0.982	0.918-1.050	0.600	42	0.088
CH_4_ intensity (g/kg FPCM)	8	0.925	0.864-0.989	0.023	19	0.278

^1^ DMI, dry matter intake; SCC, somatic cell counts; FPCM, fat and protein corrected milk; FCE, feed conversion efficiency; VFA, volatile fatty acids

**Table 4 animals-10-00620-t004:** Mean response ratio and estimated heterogeneity parameters describing the short-term effects (<28 days of treatment) of supplementing an essential oil blend (1 g/d) to dairy cows.

Parameter ^1^	*n*	Response Ratio(*R*)	95% CI	*p*-value	Heterogeneity	
			Min.–Max.		*I* ^2^	*Q*
DMI (kg/d)	17	1.000	0.976–1.024	0.988	84	<0.001
Milk yield (kg/d)	19	1.026	1.006–1.046	0.008	76	<0.001
Milk Fat (g/kg)	10	1.000	0.978–1.022	0.999	76	<0.001
Milk protein (g/kg)	10	1.002	0.991–1.012	0.731	55	0.018
Milk SCC (log/mL)	3	1.036	0.984–1.090	0.177	0	0.910
FPCM yield (kg/d)	16	1.028	1.009–1.047	0.004	69	<0.001
FCE (kg/kg)	15	1.010	0.989–1.029	0.348	40	0.055
Rumen pH	3	1.007	0.991–1.023	0.385	0	0.445
Total VFA (mmol/L)	9	0.973	0.936–1.010	0.158	4	0.400
Acetate (%)	7	1.005	0.998–1.011	0.116	22	0.260
Propionate (%)	7	1.009	0.969–1.049	0.672	39	0.131
Butyrate (%)	7	0.985	0.958–1.012	0.276	0	0.544
Protozoa (log cells/mL)	3	0.969	0.896–1.046	0.423	78	0.011
CH_4_ production (g/d)	8	0.978	0.957–0.998	0.037	0	0.675
CH_4_ yield (g/kg DMI)	8	0.980	0.923–1.039	0.497	49	0.047
CH_4_ intensity (g/kg FPCM)	7	0.974	0.944–1.003	0.087	0	0.984

^1^ DMI, dry matter intake; SCC, somatic cell counts; FPCM, fat and protein corrected milk; FCE, feed conversion efficiency; VFA, volatile fatty acids; AI, artificial inseminations.

**Table 5 animals-10-00620-t005:** Mean response ratio and estimated heterogeneity parameters describing the long term effects (≥28 days of treatment) supplementing an essential oil blend (1 g/d per cow) to dairy cows.

Parameter	*n*	Response Ratio(*R*)	95% CI	*p*-Value	Heterogeneity	
			Min.–Max.		*I* ^2^	*Q*
DMI (kg/d)	16	1.003	0.980–1.026	0.777	86	<0.001
Milk yield (kg/d)	19	1.036	1.016–1.056	<0.001	73	<0.001
Milk Fat (g/kg)	9	1.013	0.971–1.057	0.541	77	<0.001
Milk protein (g/kg)	9	0.993	0.973–1.012	0.465	88	<0.001
Milk SCC (log/mL)	3	1.000	0.987–1.012	0.972	0	0.777
FPCM yield (kg/d)	15	1.041	1.028–1.054	<0.001	5	0.392
FCE (kg/kg)	12	1.044	1.007–1.080	0.016	79	<0.001
Rumen pH	3	1.005	0.988–1.020	0.578	0	0.546
Total VFA (mmol/L)	6	0.978	0.932–1.026	0.373	5	0.383
Acetate (%)	4	1.002	0.986–1.017	0.844	0	0.494
Propionate (%)	4	1.002	0.948–1.059	0.932	0	0.994
Butyrate (%)	4	0.974	0.888–1.067	0.568	0	0.397
Protozoa (log cells/mL)	3	0.992	0.941–1.045	0.770	86	0.001
CH_4_ production (g/d)	7	0.912	0.868–0.958	<0.001	0	0.724
CH_4_ yield (g/kg DMI)	7	0.871	0.802–0.945	0.001	0	0.986
CH_4_ intensity (g/kg FPCM)	5	0.901	0.807–1.000	0.050	0	0.748

^1^ DMI, dry matter intake; SCC, somatic cell counts; FPCM, fat and protein corrected milk; FCE, feed conversion efficiency; VFA, volatile fatty acids; AI, artificial inseminations.

## Data Availability

Data is available on request to the corresponding author.

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
