# Peer review of "A Meta-analysis Describing the Effects of the Essential oils Blend Agolin Ruminant on Performance, Rumen Fermentation and Methane Emissions in Dairy Cows"

_animals, 2020, doi:10.3390/ani10040620_

Round 1

Reviewer 1 Report

Critical points in the manuscript:

  1. The title is not clear. It is a meta-analysis of agolin, a commercial product.
  2. The methodology is clear. However, I do not consider it appropriate to include unpublñished data in the analysis.
  3. There is no detailed description in agolin (this is understandable because it is a registered product). 
  4. The authors conclude that the actions of agolin is not clear. It is not adequate, the biochemical and physiological effect on the active ingredients contained in agolin is no discussed.
  5. Several reviews (Beauchemin and McAllister) indicate that oils decrease methane emissions. Therefore, the use of agolin in animals is not innovative.
  6. It is a commercial promotion to agolin. I believe the manuscript to be published in a practice journal.

Author Response

Comments and Suggestions for Authors

AU: Thank you for your comments and suggestions which have contributed to improve the quality of this manuscript.

Critical points in the manuscript:

The title is not clear. It is a meta-analysis of agolin, a commercial product.

AU: We have specified in the tittle that the blend of essential ols is ‘Agolin Ruminant’

The methodology is clear. However, I do not consider it appropriate to include unpublished data in the analysis.

AU: We consider that these unpublished studies are important because they capture the impact of this nutritional intervention under real farm conditions, which is a good complement to the trials conducted under more controlled experimental conditions.

There is no detailed description in agolin (this is understandable because it is a registered product). 

AU: A more detailed description of Agolin has been provided, L 67-69

The authors conclude that the actions of agolin is not clear. It is not adequate, the biochemical and physiological effect on the active ingredients contained in agolin is no discussed.

AU: Our meta-analysis revealed the positive productive effects of Agolin in terms of higher milk yield and lower methane emissions. Thus, our study focused on productive data. However the mode of action is not clear based on the data analysed because these studies did not assessed the rumen microbiome or metabolic pathways. This later observation justifies further studies to describe the mechanisms behind this nutritional intervention. The effects of the main active compounds within Agolin EO blend on the rumen fermentation have been described in section 4.2.

Several reviews (Beauchemin and McAllister) indicate that oils decrease methane emissions. Therefore, the use of agolin in animals is not innovative.

AU: The main problem of the use of essential oils in ruminant nutrition is the great variability in composition among products and doses. One of the key aspects of this meta-analysis was the use of the same product and dose across studies. Following your recommendation, we have indicated that the use of EO to enhance productivity and decrease enteric methane emissions is not new (Beauchemin et al., 2009, Benchaar et al., 2008), L 58-59

It is a commercial promotion to agolin. I believe the manuscript to be published in a practice journal.

AU: We do not think this paper is a commercial promotion to Agolin. This has been indicated in the “Conflict of Interest” section. We consider that this meta-analysis has been rigorously conducted using standardized methodology of analyses and the scientific expertise of colleagues from different well known and recognized research institutions and with no commercial bias. Based on this we consider it has attributes to be published in a peer-review journal

Reviewer 2 Report

Comments to authors:

This is an interesting article about the effects of a particular blend of essential oils on animal performance and productivity, rumen fermentation and methane production in dairy cows. Currently this is a very relevant topic in the field of ruminant nutrition, and the manuscript is well written. Only some minor issues should be addressed, as explained below.

  • Line 88: Please, remove “at the”, which is written twice.
  • Line 99: In Table 1 there are only 3 studies indicated with primiparous cows (not 5 as noted in the text).
  • Line 124: Looking at Table 2, it seems that there are only 7 studies with rumen fermentation and methane emissions data.
  • Line 131: This sentence is unclear: “the effect size was calculated as the ratio method”. Please, reword to make it clearer.
  • Tables 2, 3, 4 and 5: Please, correct “SSC” to “SCC” in the explanations below the table.
  • Line 152 and 178: If I have understood correctly, the effect size is calculated as the response ratio, which means that they are equivalent. If that is right, please, revise the text and try to clarify, because in some parts (e.g., abstract and line 178) it is a bit confusing.
  • Lines 158-160: You state that in some studies, the SEM was calculated as the SED divided by square root of n-1. I wonder whether this is the most appropriate way to estimate SEM. Why do you use this equation?
  • Lines 220-222: Why do you mention these numerical increases in some rumen fermentation parameters? They are not significant and, in my opinion, not relevant.
  • Lines 228-230: You have just mentioned that the number of studies analysed is small for other parameters (e.g., NEFA), so you should mention this as well for BCS, fat depth and AI required.
  • Line 243: Why is the P given in the text (0.003) different to that in Figure 3?
  • Line 257: Please, correct this sentence, because “yield” and “intensity” are swapped. Agolin affects methane intensity but not yield.
  • Line 264: Please, change “FEC” to “FCE”.
  • Lines 301-302: Why do you mention effects on the rumen microbial fermentation here? This should be explained in the next section (4.2.).
  • Line 326: Why do you allude to cinnamon oils here? They are not components of the Agolin.   
  • Line 401: Please, correct to “which are dependent on”.

Author Response

Comments to authors:

This is an interesting article about the effects of a particular blend of essential oils on animal performance and productivity, rumen fermentation and methane production in dairy cows. Currently this is a very relevant topic in the field of ruminant nutrition, and the manuscript is well written. Only some minor issues should be addressed, as explained below.

AU: Thank you for your comments and suggestions which have contributed to improve the quality of this manuscript.

Line 88: Please, remove “at the”, which is written twice.

AU: Corrected

Line 99: In Table 1 there are only 3 studies indicated with primiparous cows (not 5 as noted in the text).

AU: Corrected

Line 124: Looking at Table 2, it seems that there are only 7 studies with rumen fermentation and methane emissions data.

AU: We have double-checked the numbers and there are 8 studies reporting CH4, 8 for total VFA and 7 for VFA molar proportions. Corrections have been made accordingly.

Line 131: This sentence is unclear: “the effect size was calculated as the ratio method”. Please, reword to make it clearer.

AU: Corrected.

Tables 2, 3, 4 and 5: Please, correct “SSC” to “SCC” in the explanations below the table.

AU: Corrected

Line 152 and 178: If I have understood correctly, the effect size is calculated as the response ratio, which means that they are equivalent. If that is right, please, revise the text and try to clarify, because in some parts (e.g., abstract and line 178) it is a bit confusing.

AU: That is right. The effect size is the relationship between two variables (e.g. Raw difference in means, Response ratio, Risk ratio, Odds ratio, etc). In our meta-analysis the effect size was based on means and was calculated as a ratio (AGO divided by CTL) and defined as Response Ratio. Since the term response ratio (R) is more specific than effect size, the first has been used throughout the text.

Lines 158-160: You state that in some studies, the SEM was calculated as the SED divided by square root of n-1. I wonder whether this is the most appropriate way to estimate SEM. Why do you use this equation?

AU: Corrected. We meant SD instead of SED.

Lines 220-222: Why do you mention these numerical increases in some rumen fermentation parameters? They are not significant and, in my opinion, not relevant.

AU: This has been corrected. It has been indicated that there were no differences for rumen fermentation data.

Lines 228-230: You have just mentioned that the number of studies analysed is small for other parameters (e.g., NEFA), so you should mention this as well for BCS, fat depth and AI required.

AU: Following recommendations from reviewer 2 and 3, these data related with blood parameters, body condition and fertility have been removed from the result and discussion section given the small number of observations. These data have been moved to supplementary material.

Line 243: Why is the P given in the text (0.003) different to that in Figure 3?

AU: It has ben corrected. P=0.003

Line 257: Please, correct this sentence, because “yield” and “intensity” are swapped. Agolin affects methane intensity but not yield.

AU: Corrected

Line 264: Please, change “FEC” to “FCE”.

AU: Corrected.

Lines 301-302: Why do you mention effects on the rumen microbial fermentation here? This should be explained in the next section (4.2.).

AU: Corrected. Sentence has been moved.

Line 326: Why do you allude to cinnamon oils here? They are not components of the Agolin.

AU: This sentence has been removed.

Line 401: Please, correct to “which are dependent on”.

AU: This paragraph has been removed following reviewer 3 recommendation.

Reviewer 3 Report

The study is with the aim of the Journal and it provides using meta-analysis methodology new information useful to improve environmental sustainability of milk production. The study is well written and organized. I’ve only a concern related to the small dataset used for some variables evaluated: such as rumen pH, blood metabolites, digestibility, protozoa. The authors underlined the small dataset used for some variables and this is correct; however, for some variables such as blood metabolites, I advise not to present these results. See more detailed comments below.

Pag 5. Line 123-124. “ ..and CH4 emissions were reported in 8 studies”. In table 2, the number of studies for these variables is n=7.

Pag.5 line 127. “and in vitro incubations (studies 5 and 6). “. In the introduction the authors underline the importance of studying the rumen adaptation period to additives and the need to evaluate in vivo response. What is the reason why these in vitro studies were included? Please clarify.

Pag.5. Lines 132-133.  “Moreover, 3 studies also reported information about the health status in terms of plasma metabolites, body condition score (BCS), feed digestibility and fertility. “.  I advise to delete this part. In the following part the authors state:

Pag7. Line 191-192. “For some parameters the number of studies was small, which may represent a problem for estimating between studies variance and thus draw a firm conclusion”.

or

“The effect of Agolin on blood health parameters such as non-esterified fatty acids 227 (NEFA), calcium, gamma-glutamyl transferase (GGT) and cholesterol was negligible based on the small number of studies analysed”

Hence, as underlined also by the authors, this can be a limit in order to underline a possible effect of Agolin.

Pag 7  line 221 “but promoted a numerical increase in rumen  pH (θ = 1.006)”. The P value was not significant. Only 2 studies reported rumen pH values.

Pag 10. Lines 275-276. “… possibly because in some of the studies the TMR was offered at 95% of ad libitum to ensure uptake”. Please specify the number of studies which offered TMR at 95 % of ad libitum.

Pag 11. Lines 277-278. “These results suggest that the effect of Agolin supplementation on milk yield and milk composition may differ depending on the stage of lactation”. Do the authors have a dataset to test statistically this hypothesis?

Pag 11. 4.2 Rumen fermentation paragraph. I believe that this paragraph can be summarized.

Pag 12. Line 334-335.”.. Agolin promoted a small increment in the propionate molar proportion (+1.1%) which could favour milk yiled..” . This effect was however not significant.

Pag 12. Lines 372-380. This hypothesis is based on a really small dataset. My suggestion is reporting this computation only for the studies that present both VFA and methane.

Pag. 13. Lines 383-399. Protozoa were not statistically affected by dietary treatment; moreover, only 3 studies were used hence also this part should be interpreted very carefully.

Pag 13. 4.4 Animal’s Physiology. I advise to delete this part due to the small number of studies used.

Author Response

Comments and Suggestions for Authors

AU: Thank you for your comments and suggestions which have contributed to improve the quality of this manuscript.

The study is with the aim of the Journal and it provides using meta-analysis methodology new information useful to improve environmental sustainability of milk production. The study is well written and organized. I’ve only a concern related to the small dataset used for some variables evaluated: such as rumen pH, blood metabolites, digestibility, protozoa. The authors underlined the small dataset used for some variables and this is correct; however, for some variables such as blood metabolites, I advise not to present these results. See more detailed comments below.

AU: Following your recommendation, parameters with a low number of observations have been removed from the tables and the discussion section to prevent weak statements or speculation. They have been only kept in the supplementary material as additional information but not discussed.

Pag 5. Line 123-124. “ ..and CH4 emissions were reported in 8 studies”. In table 2, the number of studies for these variables is n=7.

AU: Numbers of studies have been double-checked throughout the paper.

Pag.5 line 127. “and in vitro incubations (studies 5 and 6). “. In the introduction the authors underline the importance of studying the rumen adaptation period to additives and the need to evaluate in vivo response. What is the reason why these in vitro studies were included? Please clarify.

AU: These studies were included due to two reasons: i) the animals were already adapted to the additive when used as rumen fluid donors, and ii) these studies were part of a large in vivo study which provided a whole dataset along with the methane emissions in vitro. Moreover, it has been reported that the use of batch culture with animals adaptaed to the diet/treatment to test can be considered as a reliable proxy method to estimate rumen CH4 emissions (Yáñez-Ruiz et al., 2016, Anim. Feed Sci and Technol., 216 1-16). Following your recommendations these aspects have been indicated in the text and we have also mentioned that this information should be carefully interpreted given this limitation, L131-138

Pag.5. Lines 132-133.  “Moreover, 3 studies also reported information about the health status in terms of plasma metabolites, body condition score (BCS), feed digestibility and fertility. “.  I advise to delete this part. In the following part the authors state:

AU: This part has been removed from the manuscript to the supplementary material and has not been further discussed.

Pag7. Line 191-192. “For some parameters the number of studies was small, which may represent a problem for estimating between studies variance and thus draw a firm conclusion”. “The effect of Agolin on blood health parameters such as non-esterified fatty acids 227 (NEFA), calcium, gamma-glutamyl transferase (GGT) and cholesterol was negligible based on the small number of studies analysed” Hence, as underlined also by the authors, this can be a limit in order to underline a possible effect of Agolin.

AU: Following your recommendations, parameters with low number of studies have been removed.

Pag 7  line 221 “but promoted a numerical increase in rumen  pH (θ = 1.006)”. The P value was not significant. Only 2 studies reported rumen pH values.

AU: It has been indicated that rumen fermentation pattern (pH, VFA, and protozoa) was not significantly affected by Agolin supplementation. Rumen pH was reported in 3 studies.

Pag 10. Lines 275-276. “… possibly because in some of the studies the TMR was offered at 95% of ad libitum to ensure uptake”. Please specify the number of studies which offered TMR at 95 % of ad libitum.

AU: It has been specified, L274

Pag 11. Lines 277-278. “These results suggest that the effect of Agolin supplementation on milk yield and milk composition may differ depending on the stage of lactation”. Do the authors have a dataset to test statistically this hypothesis?

AU: Unfortunately our dataset is not big enough to statistically demonstrate this hypothesis. As a result, we have removed this statement.

Pag 11. 4.2 Rumen fermentation paragraph. I believe that this paragraph can be summarized.

AU: this section has been summarized, L318-341

Pag 12. Line 334-335.”.. Agolin promoted a small increment in the propionate molar proportion (+1.1%) which could favour milk yiled..” . This effect was however not significant.

AU: This sentence has been removed.

Pag 12. Lines 372-380. This hypothesis is based on a really small dataset. My suggestion is reporting this computation only for the studies that present both VFA and methane.

AU: All studies which reported rumen fermentation also provided CH4 data.

Pag. 13. Lines 383-399. Protozoa were not statistically affected by dietary treatment; moreover, only 3 studies were used hence also this part should be interpreted very carefully.

AU: Following your recommendation, we have indicated that this data should be interpreted very carefully.

Pag 13. 4.4 Animal’s Physiology. I advise to delete this part due to the small number of studies used.

AU: This section has been removed following your recommendation.

Reviewer 4 Report

This is a well-written manuscript. I have very minor edits.

134 straight thought. I think you mean straight through

The first description of Agolin is in section 4.2 I suggest you move or add a description of it in the introduction

Author Response

Comments and Suggestions for Authors

This is a well-written manuscript. I have very minor edits.

AU: Thank you for your comments and suggestions which have contributed to improve the quality of this manuscript.

134 straight thought. I think you mean straight through

AU: Corrected

The first description of Agolin is in section 4.2 I suggest you move or add a description of it in the introduction

AU: The product description has been moved to the introduction as suggested.

Round 2

Reviewer 1 Report

Manuscript was improved